# Sustainable Delicacy: Variation in Quality and Sensory Aspects in Wild Boar (*Sus scrofa*) Meat and Comparison to Pork Meat—A Case Study

**DOI:** 10.3390/foods12081644

**Published:** 2023-04-14

**Authors:** Sabine Sampels, Maja Jonsson, Mats Sandgren, Anders Karlsson, Katarina Arvidsson Segerkvist

**Affiliations:** 1Department of Molecular Sciences, Swedish University of Agricultural Sciences, P.O. Box 7015, 75007 Uppsala, Sweden; 2Department of Animal Environment and Health, Swedish University of Agricultural Sciences, P.O. Box 234, 53223 Skara, Sweden

**Keywords:** carcass weight, colour, pH, n-3 fatty acids, lipids

## Abstract

The aim of this study was to evaluate quality and sensory variation in wild boar meat in comparison to pork. Meat quality in wild boar is expected to vary more compared to pork due to different feeding environment, age and gender. In order to be able to promote wild boar meat as a sustainable high-quality product, there is a need to evaluate the variation in meat quality attributes, including technological, compositional and sensory/texture aspects. We evaluated carcass characteristics, pH, colour, lipid profile and sensory aspects of wild boar meat of different age and gender and compared them with pork. Wild boars had lower carcass weight (*p* = <0.0001) and higher ultimate pH (*p* = 0.0063) compared to domestic pigs. Intramuscular fat content had a tendency to be higher in wild boar meat (*p* = 0.1010), as well as the proportion of nutritional valuable n-3 FA (*p* = 0.0029). The colour of pork was more pink (*p* = 0.0276) and pale (*p* = <0.0001) compared to meat from wild boar. Meat from wild boar gilts received the highest sensory scores. Based on these findings, we suggest that meat from younger animals could be sold in different cuts directly while meat from older animals might be more suitable for the production of sausage.

## 1. Introduction

Meat is a good source of protein of high nutritional value and an important part of the diet in many countries. In Sweden, meat consumption was estimated to be 833,420 tons for 2021, of which approximately 21,140 tons are game meat [1]. Game meat including wild boar meat is in general seen as a delicacy and a special, often local product appreciated by many consumers with an increasing demand [2,3,4]. Wennborg [5] concluded that with a more efficient supply chain, Swedish wild boar (*Sus scrofa*) meat has the potential to contribute to long-term sustainability in the food chain. This concerns especially the distribution from the hunters to the consumers. Wennborg [5] describes how a change in legislation is needed to facilitate the selling of the meat for the hunters. Other obstacles in the chain were pricing discrepancies between the game dealers and the geographical distance between dealers and hunters. Wennborg [5] highlights also that wild boar meat has a lower impact on the environment than livestock meat, also since controlling the wild boar population will lower economic losses due to reduced agricultural damages as well as reduced number of traffic accidents.

In Sweden, wild boar is an appreciated hunted game and has, since the return to southern Sweden in the late 1970s, increased markedly. The number of wild boars in Sweden has increased by a factor of over 3000% from 2000 to 2020 [6]. The exact number of wild boars is not known, so the increase of the population is measured by the number of animals shot each year. This number increased from 5204 in 2002 to 161,305 in 2020 [6]. This increase in Sweden is believed to be due to high reproduction, a change in land use towards intensification of crop production and a hunting strategy that involves not killing reproductive animals, i.e., sows [7]. A similar situation is seen all over Europe [8]. Increased hunting is one of the strategies to control and reduce the wild boar flock sizes. It is important to manage large wildlife populations by hunting in order to maintain a sustainable number of animals while at the same time preserving the species and their positive impact on biodiversity due to a balance of predators and grazers [3,9]. The Swedish government, in collaboration with the Swedish Board of Agriculture, the National Food Agency, the county administrative boards and the Swedish Veterinary Institute, started an initiative in 2020 called the Wild Boar Package, and several projects to promote wild boar meat have been started; this is in line with the Swedish national food strategy to increase consumers’ access to and supply of wild boar meat [10]. This will hopefully lead to an increased availability of meat and the need to promote it [11].

However, the variations in the quality of wild boar meat, especially sensory aspects such as colour, taste, odour and tenderness are expected to be high due to the fact that the animals have quite a variation in diet depending on the region they live in and over the year (due to different feed sources available depending on the season) [12], as well as hunting techniques used, the gender of the animals and their age at slaughter [13,14]. Mature boars could develop so-called boar taint, an unpleasant odour caused by androstenone and skatole and other compounds [15,16]. Effects on various aspects on meat quality are well studied in farmed animals, but regarding wild boar, it is largely unknown how the different aspects of feed availability and composition, age and gender and hunting technique affect sensory aspects of wild boar meat. It is also unclear which attributes customers appreciate with wild boar meat.

If it should be possible to further process the wild boar meat on a larger scale, into, for example, salami, a defined and uniform quality is needed to facilitate the production of uniform products, which is necessary for larger-scale marketing. Hence, in order to be able to promote wild boar meat as a sustainable high-quality product or use it for value-added products, there is a need to evaluate the variation in meat quality within technological, compositional and sensory/texture aspects and also, what extent of variation consumers can accept. The aim of this study was therefore to evaluate central quality and sensory aspects in wild boar meat in comparison to pork.

## 2. Material and Methods

### 2.1. Sample Collection

For evaluation of technological meat quality, shoulder muscle (*M. triceps brachii*) from 17 wild boars was obtained at the end of October 2021 from a game handling facility in the south of Sweden. Additional muscles from 3 domestic pigs were obtained from a commercial abattoir. The gender of these animals was not noted as no variation was expected and they should mirror supermarket situation.

The wild boars were both females and males of varying ages, randomly selected from what was hunted the week before the sampling. Meat was stored at −20 °C until used for analysis.

For sensory evaluation, samples from 3 wild boars (1 male, 1 sow and 1 gilt) obtained in January 2022 from the game handling facility and from 1 domestic pig from the same commercial abattoir were used. The carcasses from the wild boars and pigs were weighed at the game handling facility and the abattoir, respectively.

### 2.2. pH and Colour Measurement

The pH (ultimate pH) was measured at three positions on the shoulder muscle (*M. triceps brachii*) of each animal after thawing at 4 °C for 24 h, followed by equilibration for one hour at room temperature, using a 340 WTW pH-meter (Weilheim, Germany). Calibration of the instrument was performed with standard pH buffers of 4.00 and 7.00 at 20 °C.

For the colour measurement, the meat was let to bloom at room temperature for one hour before the measurements which were performed in triplicates using a chroma meter (CR-300 Chroma Meter, Minolta, Japan) according to the method of [17]. The instrument was calibrated before each measurement against a white tile (L* = 93.30, a* = 0.32 and b* = 0.33).

### 2.3. Lipid Analyses

For lipid analysis, the entire shoulder muscle (*M. triceps brachii*) from each sampled animal was grounded, and 1 g was used for lipid extraction, according to [18]. Total lipid content was quantified gravimetrically. For fatty acid (FA) profile analyses, methylation of total lipids was performed according to Appelqvist [19] with BF_3_. Methylated FA were analysed by gas chromatography (GC) using a Trace Ultra FID (Varian AB, Stockholm, Sweden) using a BPX-70 50 m fused silica capillary column (id. 0.22 mm, 0.25 μm film thickness, SGE, USA) as described earlier [20]. The peaks were identified by comparing sample retention times to retention times of the standard mixture GLC-68-D (Nu-Chek Prep, Elysian, USA).

### 2.4. Sensory Evaluation

For the sensory test, the whole shoulder muscles (*M. triceps brachii*) from wild boars and pigs were minced, no salt or flavour added, vacuum-packed, and heated in a water bath at 70 °C until the core temperature was 67 °C [17]. The samples were served to the panellists on paper plates. Crackers and water were available to clean the palette.

The first part of the evaluation was carried out as a preference test comparing 2 samples in 6 combinations. Parameters evaluated were colour, aroma, taste, tenderness, and juiciness. In a second step, overall acceptability for each of the four samples was evaluated using a scale consisting of 7 points, where 1 corresponded to “dislike” and 7 corresponded to “like most” [21]. The panel consisted of ten randomly invited persons (5 men and 5 women) from staff and students at the Department of Molecular Science at the Swedish University of Agricultural Sciences (SLU) Ultuna, Uppsala. Choice for an equal number of men and women was done because it has been shown earlier that women are more sensitive to boar taint [22]. The low number of panellists was due to the available amount of meat; we considered this also as a pilot study to confirm the suitability of the method.

### 2.5. Statistical Analysis

Data from pH, color and lipid content were analyzed with the Mixed procedure in SAS (SAS 9.4, SAS Inst. Inc., Cary, NC, USA), A general Satterthwaite approximation for the denominator degrees of freedom was performed, using the SATTERTH option in SAS. Differences were considered significant at *p* < 0.05 and as a tendency for significance at 0.05 < *p* < 0.10. Fatty acids and lipid classes are presented as mean and standard error, respectively.

The sensory test was analyzed with one-way ANOVA and General Linear Model using Minitab version 19.2020.2.0. Tukey’s test was used to make pairwise comparisons with a difference considered significant if it was *p* < 0.05.

## 3. Results and Discussion

### 3.1. Carcass Characteristics

Carcasses of domestic pigs had a significantly higher weight than wild boars, but there was no difference in carcass weight between the sexes of the wild boars (Table 1). The carcass weights for the wild boars varied from 16 kg to 51 kg. The reason for the large variation in weight for wild boars can be due to such factors as age, availability of feed and individual variations. Ludwiczak et al. [23] also found a similar variation in juveniles and sub-adult wild boars between 24 and 60 kg measured as dressed field weight, and Zmijewski & Modzelewska-Kapitula [13] found a variation from 18 to 62 kg between 1- and 3-year-old animals. The pigs, on the other hand, are produced under controlled conditions, were of the same breed and were slaughtered at about the same age and weight, which gives a more homogeneous carcass weight. The average carcass weight of 92.7 kg is also similar of typical slaughter weights of commercial pigs in Europe [24].

### 3.2. pH and Colour

The ultimate pH of wild boar was significantly higher than in domestic pig and varied from 5.55 to 5.87, with an average of pH 5.70 and 5.64 for boars and sows, respectively. The final pH of meat from domestic pig ranged from 5.48 to 5.52 with an average pH of 5.50. There was no difference between the pH in meat from male and female wild boars (Table 1). Ludwiczak et al. [23] found an ultimate pH in M semimembranous for wild boars ranging from 5.45 to 5.88 and similarly to our results, no difference between male and female animals. Fernanda & de Felício [25] found slightly lower pH 48 h values of 5.47 for wild boar and 5.34 for pig, but similarly to our values, higher ultimate pH in wild boar meat. The higher pH in wild boar meat compared to pork is most probably a result of higher activity of the animals before slaughter, resulting in lower amounts of available glycogen [26]. The higher pH in the wild boar meat could lead to shorter shelf life but results probably also in a higher water holding capacity. This could be advantageous for certain products such as, for example, emulsion sausages, but less suitable for fermented products where a low pH is required in the final product as well as having a low water activity (a_w_) [27,28].

The meat from domestic pigs had a much lighter and more pinkish colour compared to the wild boar meat. This was reflected in significantly higher lightness (L*) and redness (b*) values and significantly lower yellowness (a*) values. Ivanovic et al. [29] found similar results. In addition, for this trait, there was no difference in colour between the meat from male and female wild boars. Fernanda & de Felício [25] found also a higher L*-values in commercial pig compared to wild boar from Brazil, but the L*-values they measured were in general higher compared to our values, meaning the meat was lighter. Lebret et al. [26] showed also a correlation between lower L* and higher pH in meat from French local Basque pigs compared to conventional large white pigs. They also found a higher redness in the Basque pigs and suggested that breed difference was the reason for this, which could also be true for the difference between wild boars and domestic pigs. This hypothesis is confirmed by the work from [29], where meat from a crossbreed of Duroc, Yorkshire and wild boar showed intermediate values in comparison to the meat from Duroc, Yorkshire and pure wild boar. In contrast to our results, Fernanda & de Felício [25] found lower a*-values for wild boar meat compared to meat from commercial pigs and vice versa for b*-values. These differences to our study are most probably due to differences in feed composition and physical activity. Higher contents of carotenoids, for example, could increase yellowness in the meat and more xanthophyls decrease redness, while higher activity will increase red muscle fibre proportion and myoglobin content and thereby, increase the muscles’ redness [30,31]. As corn rich in zeaxanthin is used for bait for wild boars in Sweden, this could be the reason for the higher b*-values in our study. For example, An et al. [32] found higher redness in pork from animals fed palm kernel meal, but on the other hand, Citek et al. [33] did not find a difference in yellowness in pork from pigs fed different ratios of corn. Another factor creating a difference in colour could be fatness, as higher fat content gives a lighter meat [30]. A possible variation of wild boar meat colour in relation to factors like stress, physical activity, feed composition and feed levels needs to be explored more.

### 3.3. Fat Content and Fatty Acid Composition

Regarding fat content, wild boars had a tendency towards higher muscle fat content with 4.7% in wild boar versus 2.9% in pig (*p* = 0.101; Table 2). Fatty acid composition showed significantly higher amounts of n-3 FA in wild boar and even small proportions of the long chain n-3 polyunsaturated FA (PUFA) 20:5 n-3 and 22:5 n-3, but this was not detected in meat from domestic pig. In wild boar meat, the proportion of 18:3 n-3, which is abundant in grass and leaves, is in line with the findings of Pedrazzoli et al. [34], where higher proportions of 18:3 n-3 in wild boar meat from forests were discussed. The long chain PUFA 20:5 n-3 and 22:5 n-3 can be partly synthetized by the animals if they have enough 18:3 n-3 in their diet. These PUFA could also come naturally from the diet as, for example, lichen can contain those long chain PUFA [35]. These PUFA are considered as nutritionally valuable and important in the human diet [36].

On the other hand, the wild boars included in our study also showed substantially higher proportions of 18:2 n-6, probably due to bait feeding with corn which is rich in this FA [37], which is, however, less favourable from a human nutritional point of view [36]. In contrast to our results, Ivanovic et al. [29] found lower values of 18:2 n-6 in wild boars compared to pork. In their study, wild boars came from the Serbian steppe with forest vegetation predominating by oak, elm, linden, chestnut and hazel trees and herbaceous species as *Graminaceae*, *Astecraceae* and *Poaceae*. This is a similar feed composition as described by Pedrazzoli et al. [34]. As mentioned earlier, green pasture is naturally higher in 18:3 n-3, also resulting in meat with higher contents of n-3 FA [38]. This supports our hypothesis that the corn bait might be responsible for the high 18:2 n-6 content in wild boar meat in the present study.

Still, the ratio n-6/n-3 was lower in wild boar meat than in meat from pigs (12.7 and 10.9 versus 15.7 in wild boar and domestic pig, respectively), indicating a higher nutritional value of wild boar meat compared to pork. However, this difference was not significant due to a high variation in the pig. In general, a diet rich in n-3 FA is considered more healthy and the recommended ratio between n-6 and n-3 FA in the diet is about 4 or below [36,39].

No significant difference between male and female wild boars muscle FA composition was found. Razmaite et al. [40] also found the long chain n-3 PUFA in meat from wild boars in Lithuania and also, relatively high proportions of 18:2 n-6 and no difference between the sex of animals. For 18:2 n-6, they found a significant seasonal variation, with higher values in winter as was unsaturation in general [40]. The authors connect differences in n-3 and n-6 contents to availability of feed and difference of feed in forest and farmlands but also suggest a thermoregulatory adjustment of the animals for the need of a higher unsaturation in the fat during winter. In the present study, we only analysed one sampling point and there is a need to analyse meat from animals shot at different seasons as well as from different environments and feed backgrounds to be able to draw final conclusions and give nutritional recommendations for Swedish wild boar meat.

### 3.4. Sensory

The results of the sensory evaluation of meat from wild boar gilts showed that this meat was preferred compared to meat from pigs (Table 3). Wild boar gilt received the highest grade with a mean value of 5.9 ± 1.2 for ‘overall liking’, compared to 3.9 ± 1.6 for meat from pigs. Wild boar sow had intermediate scores, while wild boar male had similar scores as meat from pig. Colour impression values were highest for wild boar gilts and lowest for meat from pig, while wild boar male and sow had similar intermediate values that also differ significantly from both wild boar gilt and pig. For the other parameters, aroma, taste, tenderness and juiciness, no differences were found by the panellists. The panellists were also allowed to comment on the samples. The pork sample received three comments that it was dry and pale, which can be perceived as negative or unappetizing properties. The sample of entire wild boar male meat received no comments, suggesting no detectable off flavours due to boar taint. A drawback of the study is the small number of recruited panellists, which was mainly due to the small amounts of wild boar meat and some difficulties during sampling. Hence, we decided to do this test as a pilot study and the results should be treated as preliminary. Further sensory evaluation also on a broader variation of animals is necessary for final conclusions. However, we believe the results can indicate a general trend.

There are not many studies comparing wild boar and pig meat. However, Ivanovic et al. [29] compared meat from crosses of Duroc x Yorkshire, Duroc x Yorkshire x wild boar, and wild boar, and pure wild boar received the highest grades for flavour.

Our results suggest that meat from younger wild boars is more appreciated by the consumers. This could result in focusing on selling whole meat cuts from younger animals, while using meat from older animals for processing into different kinds of sausages and pâtés, for example. However, as mentioned above, it is important beside sex differences to study also how other factors such as different regions and seasons might influence meat quality attributes before final conclusions can be drawn. In addition, to get a broader picture, there is also a need to execute sensory evaluation of different whole meat cuts in order to evaluate tenderness in relation to the above-mentioned factors that can affect meat quality. Jukna & Valaitiene [41] found a higher tenderness in pig meat compared to wild boar using Warner Brazler Shearforce while, on the other hand, Ivanovic et al. [29] found higher tenderness in wild boar compared to pork in a sensory evaluation. Possible differences need to be evaluated for the Swedish population and conditions.

## 4. Conclusions

More analyses are needed but the present results indicate an effect of the FA composition of the used bait (most probably corn) on the meat composition of wild boar resulting in relatively high 18:2 n-6 contents. It would be worth considering a change of bait to a crop with lower n-6, such as, for example, rapeseed cake. The higher pH of wild boar meat might reduce its shelf life due to a risk for faster microbial growth but could also be advantageous for processing with a focus on emulsion sausages. Meat from younger animals could be sold in different cuts directly while meat from older animals might be better to produce sausages and pâtés, for example. No differences were found when comparing the same parameters between male and female animals from wild boar, indicating no need to have separate processing or marketing strategies for meat from animals of a different gender. However, in this study, only animals from one time point, one hunting method and from one location were sampled and there is therefore a need for further evaluation of meat quality and consumer preferences on a larger scale to be able to estimate the variation and suggest marketing strategies. In addition, other quality parameters such as water holding capacity and drip loss at different stages need to be further evaluated. Besides, this consumer attitude towards game meat and hunting is an important factor to evaluate before meat can be marketed on a bigger scale.

## Figures and Tables

**Table 1 foods-12-01644-t001:** Technological quality traits for wild boar and pig. Data presented as means and standard deviation.

Trait	Entire Wild Boar Male (*n* = 9) ⸸	Wild Boar Sow (*n* = 8)	Pig (*n* = 3)	*p*-Value
Carcass weight (kg)	32.0 ± 3.25 ^b^	31.7 ± 3.44 ^b^	92.7 ± 5.62 ^a^	<0.0001
Ultimate pH	5.70 ± 0.03 ^a^	5.64 ± 0.03 ^a^	5.50 ± 0.05 ^b^	0.0063
L*	32.3 ± 1.09 ^b^	31.3 ± 0.94 ^b^	47.2 ± 1.54 ^a^	<0.0001
a*	15.2 ± 0.88 ^a^	15.4 ± 0.76 ^a^	2.36 ± 1.24 ^b^	<0.0001
b*	19.3 ± 0.77 ^b^	18.9 ± 0.66 ^b^	22.7 ± 1.08 ^a^	0.0276

**⸸** Entire wild boar male: *n* = 9 for weight and pH, *n* = 6 for colour; Abbreviations L* = lightness; a* = redness; b* = yellowness; Different superscript letters in a row indicate significant difference (*p* < 0.05).

**Table 2 foods-12-01644-t002:** Total lipid content (g/100 g tissue) and individual fatty acid content (g/100 g total identified FA) in meat from wild boar and pig. Data are given as means and standard deviation.

Fatty Acid	Entire Wild Boar Male (*n* = 9)	Wild Boar Sow (*n* = 8)	Pig (*n* = 3)	*p*-Value
Total lipid	4.08 ± 0.58	5.42 ± 0.61	2.92 ± 1.00	0.1010
14:0	1.11 ± 0.13	1.30 ± 0.14	1.23 ± 0.22	0.6016
16:0	23.1 ± 0.49	24.1 ± 0.51	25.3 ± 2.84	0.1055
16:1 n-9	0.40 ± 0.10	0.47 ± 0.10	0.31 ± 0.17	0.7155
16:1 n-7	2.35 ± 0.26 ^b^	2.81 ± 0.27 ^b^	4.13 ± 0.45 ^a^	0.0113
17:0	0.18 ± 0.08	0.28 ± 0.08	0	0.2209
18:0	10.3 ± 0.37	10.3 ± 0.39	10.7 ± 0.64	0.8203
18: 1n-9	35.0 ± 1.48	36.9 ± 1.57	42.2 ± 2.57	0.0782
18:1 n-7	4.14 ± 0.15	3.82 ± 0.16	4.38 ± 0.27	0.1659
18:2 n-6	16.8 ± 1.05 ^a^	15.2 ± 1.11 ^a^	8.69 ± 1.81 ^b^	0.0045
18:3 n-3	0.90 ± 0.12 ^a^	1.07 ± 0.12 ^a^	0.31 ± 0.20 ^b^	0.0172
20:1 n-9	0.58 ± 0.10	0.34 ± 0.10	0.52 ± 0.17	0.2663
20:2 n-6	0.47 ± 0.08 ^a^	0.34 ± 0.08 ^a^	0 ^b^	0.0230
20:3 n-6	0.33 ± 0.12	0.26 ± 0.13	0.17 ± 0.21	0.7877
20:4 n-6	3.37 ± 0.52	2.24 ± 0.55	2.12 ± 0.91	0.2790
20:5 n-3	0.44 ± 0.08	0.29 ± 0.09	0	0.0520
22:5 n-3	0.61 ± 0.12	0.33 ± 0.13	0	0.0548
SFA	24.3 ± 0.56	25.4 ± 0.59	26.5 ± 0.97	0.1346
MUFA	42.3 ± 1.38 ^a^	44.4 ± 1.47 ^a^	51.5 ± 2.39 ^b^	0.0143
PUFA	5.66 ± 0.73	4.18 ± 0.78	2.60 ± 1.27	0.1183
n-6	21.0 ± 1.59 ^a^	18.1 ± 1.68 ^a^	11.0 ± 2.75 ^b^	0.0199
n-3	1.96 ± 0.20 ^a^	1.69 ± 0.22 ^a^	0.31 ± 0.35 ^b^	0.0029
n-6/n-3	12.7 ± 2.45	10.9 ± 2.59	15.7 ± 4.24	0.6322

Abbreviations: SFA: saturated fatty acids, MUFA: monounsaturated fatty acids, PUFA: polyunsaturated fatty acids. Different superscript letters in a row indicate significant difference (*p* < 0.05).

**Table 3 foods-12-01644-t003:** Sensory results from pig meat and different types of wild boar meat by untrained panelists (*n* = 10).

	Pig	Wild Boar Male	Wild Boar Sow	Wild Boar Gilt	*p*-Value
Overall liking	3.9 ± 1.6 ^b^	4.3 ± 2.0 ^b^	4.8 ± 1.14 ^ab^	5.9 ± 1.2 ^a^	0.031
Colour	0.1 ± 0.3 ^a^	1.4 ± 0.8 ^b^	1.0 ± 0.9 ^b^	2.3 ± 0.5 ^c^	<0.0001
Aroma	1.2 ± 1.0 ^a^	0.5 ± 0.9 ^a^	0.7 ± 0.8 ^a^	0.6 ± 0.7 ^a^	0.287
Taste	0.6 ± 0.7 ^a^	1.6 ± 1.2 ^a^	0.7 ± 0.8 ^a^	1.8 ± 1.3 ^a^	0.025
Tenderness	0.9 ± 1.0 ^a^	0.7 ± 0.7 ^a^	0.9 ± 0.7 ^a^	1.0 ± 0.8 ^a^	0.869
Juiciness	1.6 ± 1.2 ^a^	0.9 ± 0.6 ^a^	0.9 ± 0.6 ^a^	1.0 ± 1.1 ^a^	0.245

Overall liking of pork and wild boar meat in points on a scale of 1 to 7, where 1 is lowest and 7 is highest. For the parameters colour, aroma, taste, tenderness, juiciness, all samples were set against each other in pairs of two and the score is an average of how many times the sample was preferred over the compared sample. Different superscript letters in a row indicate significant difference (*p* < 0.05).

## Data Availability

Data available on request. The data presented in this study are available on request from the corresponding author.

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
