# Peer review of "Sustainable Delicacy: Variation in Quality and Sensory Aspects in Wild Boar (Sus scrofa) Meat and Comparison to Pork Meat—A Case Study"

_foods, 2023, doi:10.3390/foods12081644_

Round 1
Reviewer 1 Report
The manuscript presents some interesting findings and recommendations for the utilization of wild boar meat to ensure sustainability and diversify meat source. The language is clear and easy to understand. The hypothesis is sound and clear.
· In the discussion, authors have discussed on the age of wild boars and their effects. But in the Results, no age was mentioned (I agree with the authors as it is very challenging), so better focus on other aspects.
· Authors presented the data of wild sows also; so better to edit the title accordingly.
Other comments are as follows-
i. L15: lipid content and composition could be written as lipid profile
ii. L16-19: please indicate p-value
iii. L20-22: please first mention the age-wise effect based on which this conclusion has been derived. L 20-22 is true on the basis of meat quality but in this study, it is not studied.
iv. L50: please detail the positive impact.
v. L66: costumer may be customer
vi. L80: pH fall is very prominent during first 24 h. Collection of muscle were done immediately after hunting? L82-83: after harvesting it is stored at -80 degree; so pH24h may differ than the actual value. Please check this aspect.
vii. Sensory evaluation: please describe the criteria for selecting panellists and presentation of the samples
viii. Table 1: final pH is ultimate pH?
ix. Results and discussion: Appropriate
Author Response
Thank you for the possibility to revise our manuscript. We addressed all comments and hope that the manuscript can be publish in the revised form. Please find our response to the specific comments below. All changes are made with 'Track changes' in the manuscript.
Sincerely yours, in the name of all authors, Sabine Sampels
Reviewer 1:
Age of the animals: We do not know the exact age of the wild animals, however we can distinguish between adult animals which we call boars and sows and not sexually mature animals (gilts) which are younger. The judgment was done based on weight and development of teeth and we choose carefully from the animals that were delivered in the sampling week (as described in the methods section). Hence, we still think we can discuss this.
Regarding wild boar sows: The term ‘wild boar’ is both used for the species (Sus scrofa) and for the male animals in the species, so the title is conclusive. To be more clear we added the latin name in the title. We also added it at the first time the term shows up in the text.
- L15: lipid content and composition could be written as lipid profile
has been changed
- L16-19: please indicate p-value
has been added
iii. L20-22: please first mention the age-wise effect based on which this conclusion has been derived. L 20-22 is true on the basis of meat quality but in this study, it is not studied.
We studied sensory in animals of different age and it is described here ‘gilts’ is the technical name for young not sexually mature animals, please see also my explanation above regarding age determination.
- L50: please detail the positive impact.
We added that there needs to be a good balance between different species.
- L66: costumer may be customer
Corrected
- L80: pH fall is very prominent during first 24 h. Collection of muscle were done immediately after hunting? L82-83: after harvesting it is stored at -80 degree; so pH24h may differ than the actual value. Please check this aspect.
As stated in the text, pH was not measured at the game handling facility but on thawed meat. Before thawing the meat was stored in -20 not in -80°C (see text). We measure final (ultimate) pH not pH 24 as this was not possible due to the fact that we had to take the carcasses from the game handler. (Thawing time was 24h). But this we also state in the description. We added the word ultimate pH for clarification in the text. We found a mistake in the abstract were it said pH was measured after 24h we deleted that.
- Sensory evaluation: please describe the criteria for selecting panellists and presentation of the samples
As described in the section “Sensory evaluation” panellists were randomly chosen with the only set parameter to have equal number of men and women as women have shown to have a higher sensitivity towards boar taint. We added a comment and reference in the text.
- Table 1: final pH is ultimate pH?
We changed the wording
Reviewer 2 Report
General comments:
The manuscript deals with the evaluation of wild boar meat and fat quality, and its comparison to domestic pig. In my opinion, the topic addressed in the presented manuscript is highly relevant for animal-based food production and falls into the scope of the journal “Foods”. Namely, recently a lot of attention is given to alternative food sources, especially of local origin. This study addresses two important issues at the same time, food autonomy and a manner to reduce the wild boar population, a species that makes substantial damage to agriculture and the natural environment due to overpopulation.
In general, the manuscript is very well written and organised. The main concern is the low number of repetitions for domestic pigs (n=3) which should be explained, as well as the fact why the authors did not choose to collect samples of meat and fat from a female and male domestic pig. Nevertheless, the results are properly discussed and compared to relevant literature. I suggest that the explanation that this is only a case study is added already in the title of the manuscript. Also, it would be good to add the estimated age of the animals evaluated in each group (domestic pig, wild boar, wild gilt, wild sow) to the tables and text. Additionally, for wild boars (males) boar taint compounds concentration in fat tissue would be very informative, to explain if this is the reason for the higher liking of female wild pigs although authors claim that no comments were received by panelists regarding off-flavour.
I recommend a minor revision of the manuscript in its present form and suggest the following changes.
Specific comments:
Title
I suggest changing to: “A case study of sustainable delicacy - Variation in quality and sensory attributes in pork from wild boar and its comparison to domestic pig”
Abstract:
Line 10: a space is missing between “Abstract:” and “The”, also “The” is in bold
Line 12: I suggest changing to “different feeding environment, age, and gender.”
Line 21-22: I suggest changing to “might be more suitable for the production of sausage”
Introduction
Line 25: I suggest deleting “very”
Material and methods
Line 80: a space is missing between “2021:” and “from”
Line 81: please explain why only 3 samples were collected and of which sex
Line 83: Please explain “varying ages”
Line 90: large spacing between lines should be deleted
Line 101: large spacing between lines should be deleted
Line 113: large spacing between lines should be deleted
Line 126 large spacing between lines should be deleted
Line 131-132: please change “P” to “p”
Line 137 large spacing between lines should be deleted
Author Response
Thank you for the possibility to revise our manuscript. We addressed all comments and hope that the manuscript can be publish in the revised form. Please find our response to the specific comments below. All changes are made with Track change in the manuscript.
Sincerely yours, in the name of all authors, Sabine Sampels
Reviewer 2
Low number of domestic pigs: Thank you for the comment, we intended only to have an overall comparison, to what consumers get on the marked, so case study is probably a good description. We didn’t expect any variation between male and females or different age as conventional slaughtered animals are quite young, and all the same age (6-7 month).
Age: it is not really possible to determine the age of the wild animals correctly, we can just approximate by slaughter weight and teeth, so we choose not to write numbers as this would not be correct in any way.
Boar taint: as none of the panellist commented on boar taint we decided to not do these analyses, but the comment is valid and we will keep this in mind for the next time.
Title
I suggest changing to: “A case study of sustainable delicacy - Variation in quality and sensory attributes in pork from wild boar and its comparison to domestic pig”
Thank you for the suggestion, we suggest a variation: “Sustainable delicacy: Variation in quality and sensory attributes in pork from wild boar and its comparison to domestic pig – a case study”
Abstract:
Line 10: a space is missing between “Abstract:” and “The”, also “The” is in bold
changed
Line 12: I suggest changing to “different feeding environment, age, and gender.”
changed
Line 21-22: I suggest changing to “might be more suitable for the production of sausage”
changed
Introduction
Line 25: I suggest deleting “very”
changed
Material and methods
Line 80: a space is missing between “2021:” and “from”
corrected
Line 81: please explain why only 3 samples were collected and of which sex
We didn’t note the gender of the animals as there should not be any difference as explained earlier and they were just thought to mirror the general quality at the supermarket. (but we needed that specific muscle to compare similar things so we chosed to take samples directly from slaughterhouse and not from a supermarket.
Line 83: Please explain “varying ages”
Animals had different age as we had to take what was available, in general there were more younger animals so that was easy to get but the older animals did vary in age
Line 90: large spacing between lines should be deleted
Line 101: large spacing between lines should be deleted
Line 113: large spacing between lines should be deleted
Line 126 large spacing between lines should be deleted
I think these are type-setting issues but I tried
Line 131-132: please change “P” to “p”
done
Line 137 large spacing between lines should be deleted
See comment above